# Role of occupational therapy in reducing and managing violence among mental health inpatients: a scoping review protocol

David Bell ![ORCID],[1,2] Nutmeg Hallett[1]

[1]School of Nursing, University of Birmingham, Birmingham, UK
[2]Occupational Therapy Department, St Andrew's Healthcare, Birmingham, UK

**Correspondence to**
David Bell;
dsbell@standrew.co.uk

## ABSTRACT

**Introduction** Violence is pervasive among psychiatric inpatients and has profound consequences for its victims, its perpetrators and mental health services. Currently, the unique contribution of occupational therapists to reducing and managing violence among this patient group has not been systematically explored. However, an a priori model which provides an initial understanding of its role in this respect can be identified from the wider scholarly literature. This scoping review aims to apply and refine this model, thereby producing an amended version that will form the basis for further research.

**Methods and analysis** This scoping review is based upon guidance from the Joanna Briggs Institute, Levac, Colquhoun and O'Brien's scoping review framework, and the Preferred Reporting Items for Systematic Reviews and Meta-Analyses Scoping Review checklist. Electronic databases (Cumulative Index to Nursing and Allied Health Literature (CINAHL) Plus, PsycINFO, Medline, PsycARTICLES, ProQuest Health and Medicine, Allied and Complementary Medicine Database (AMED) and Google Scholar) and grey literature will be searched to identify relevant papers. Included articles will apply occupational therapy theory or occupational science to the reduction or management of violence among psychiatric inpatients and will be critically appraised by two independent reviewers. Study characteristics will be presented using frequency counts, and qualitative data will be analysed using 'best-fit' framework synthesis and secondary thematic analysis to produce an overall model of occupational therapy's contribution to violence management and reduction.

**Ethics and dissemination** Results will be disseminated through a peer-reviewed academic journal and via professional conferences. The review will collect secondary data and therefore will not require ethical approval.

## INTRODUCTION

Violence may be defined as 'an act [committed] with some degree of wilfulness that caused/has the potential to cause physical or serious psychological harm to another person/persons'.[1] Within mental health inpatient settings, violent acts are both chronic and pervasive. Indeed, a recent systematic review of patient safety within these settings

## Strengths and limitations of this study

► This study is the first attempt to map out the unique role of occupational therapy in reducing and managing violence in psychiatric inpatient settings.

► Using a combination of 'best-fit' framework synthesis and secondary thematic analysis will provide a model that organises and makes explicit existing knowledge in this respect.

► Excluding self-harm as an act of violence may result in relevant studies being omitted from the final review.

reported that 43% of participants (staff and patients) had suffered physical violence and 57% verbal aggression.[2]

While evidence from systematic reviews suggests that psychiatric inpatients who commit acts of aggression and violence are most likely to be younger, men and involuntarily admitted with a diagnosis of schizophrenia,[3 4] other studies report high rates of violence among entire psychiatric populations. For example, Iozzino *et al*'s[5] meta-analysis reported that nearly 20% of psychiatric acute inpatients display acts of physical violence during their treatment in hospital. Similarly, Broderick *et al*[6] reported that 31% of patients in a large, multihospital forensic system committed at least one physically violent assault during their hospitalisation. Furthermore, additional studies have shown that violence is a common feature of psychiatric inpatient settings, irrespective of country,[2 5 7] patients' gender,[7 8] patients' diagnosis[7 8] and civil or forensic status.[2 8] Thus, while psychiatric inpatients who act aggressively and violently may commonly fit a particular demographic, violence is by no means the preserve of a small group of psychiatric inpatients.

Given its prevalence, violence in psychiatric inpatient settings inevitably has severe and

far-reaching consequences. This is most obviously the case for its victims, who have reported increased physical health problems, decreased emotional well-being and various psychological impairments, most commonly symptoms of post-traumatic stress disorder.[9] While healthcare professionals and patients can both be affected directly in this manner, patients experience further suffering through the reduction in the quality of care they receive from assaulted healthcare workers.[9]

Additionally, the perpetrators of violence experience negative outcomes. A recent systematic review concluded that there is some evidence to suggest that agitated and aggressive psychiatric inpatients experience longer periods of admission, increased probability of readmission and increased medication usage.[10] Moreover, they are likely to be affected by restrictive practices, for example seclusion, or chemical or physical restraint, which are used to contain violence and aggression. These interventions are associated with post-traumatic stress disorder in 25%–47% of cases, as well as predominantly negative emotions, including distress and feeling punished.[11]

Finally, violence has negative consequences for healthcare services. Practices to contain violence entail significant financial costs, calculated at €822 per episode.[12] Additionally, patient aggression has been correlated with burnout in forensic psychiatric nurses,[13] higher levels of work stress among forensic nurses,[14] and decreased job satisfaction among healthcare workers.[9] Such factors inevitably increase the financial burden on services, through reduced effectiveness and retention of staff.

The prevalence and impact of violence among psychiatric inpatients make it unsurprising that considerable efforts have been made to understand its causal factors. Historically, the dominance of the medical approach within healthcare services has meant that violence has been understood as having pathological origins,[15] with its roots in patients' mental illness and personality structures. In contrast, more recent understandings have emphasised the importance of environmental factors and more specifically, their complex interactions with these patient-related factors.[15 16] Thus, recent efforts to understand violence in psychiatric settings have been far more diverse in their scope, and have sought to identify how violence is influenced by (among other factors), the physical environments of wards,[17] nurse and patient interpersonal styles,[18] ward social climate and sense of community,[19] and coercive measures.[15] Perhaps most comprehensive in this respect is Cutcliffe's and Riahi's[15 16] systemic model of variables that contribute to aggression and violence, which incorporates phenomena related to the environment, individual patients, individual clinicians and a mental healthcare system in its entirety. Similarly, Nijman[20] presents a model that identifies patient, ward and environmental factors that can cause violence; however, this model outlines more explicitly the interplay between these factors, and the 'vicious circle' whereby they interact not only to cause violence, but to reinforce it. Both of these models highlight the shift away from using a medical approach to understand violence, towards using a biopsychosocial approach.

Guidelines from the National Institute for Health and Care Excellence[21] reflect the complexity of these causal factors, and highlight the importance of adopting a multidisciplinary approach to the assessment and management of violence within inpatient psychiatric settings. To date, the unique contribution of occupational therapists in this respect has not been systematically explored. This is surprising, since they are core members of multidisciplinary teams within these settings, and use rich and diverse theoretical frameworks[22] to promote individuals' ability to perform meaningful occupations. These patterns of meaningful occupations are a central part of recovery from mental illness, due to their importance for providing an individual with a sense of purpose and a clearly defined personal identity, both of which enable participation in society.[23] Since this process of recovery from mental illness is undermined by violence, its reduction must be a key concern for occupational therapists. Despite the lack of evidence regarding their contribution in this respect, the wider literature concerning occupational therapy and violence risk assessment suggests three main ways in which occupational therapists are vital to reducing and managing violence.

First, tools that are used to assess the risk of violence and/or recidivism incorporate risk and protective factors that are key areas of interest for occupational therapists. For example, the HCR-20[1] includes problems with employment and future living situation as risk factors for violence. These domains relate to core areas of interest for occupational therapists, namely an individual's capacity for work/education and their ability to live independently. Similarly, the Structured Assessment of Protective Factors[24] includes other core occupational therapy concerns, for instance leisure activities, regular paid or unpaid work, and financial management. Such tools highlight that occupational therapists' efforts to develop individuals' skills for independent living, for meaningful education/employment and for satisfying leisure pursuits contribute directly to minimising violence risk factors and maximising protective factors. Although these tools may be used more commonly in forensic mental health settings, they do highlight the relationship between meaningful occupations and violence, irrespective of the civil or forensic background of patients. Indeed, the role of occupational therapists in assessing contextual factors for violence risk has been recognised by Blank,[25] who argued that due to their expertise in this area, occupational therapists should be core contributors to violence risk assessment tools.

Second, occupational therapy theory identifies the importance of occupations for shaping a person's sense of identity. This is perhaps most apparent in Kielhofner's[26] idea of 'occupational identity', and Wilcock's mantra of 'doing, being and becoming'.[27] More recently, Twinley[28] has coined the phrase 'dark occupations' and explored the ways in which individuals engage in activities

that, although harmful to oneself or others, can provide a sense of meaning, purpose, or identity. Indeed, she has argued that violence itself can be a meaningful occupation for individuals.[29] In contrast, Aldrich and White[30] have contested the idea of violence as a meaningful occupation, and rather frame it as an instrument of occupations and occupational identities (eg, being a career criminal). Irrespective of whether violence is an occupation in itself, or an instrument associated with other occupations, understanding how it relates to a person's occupational identity cements the role of occupational therapists in providing alternative, prosocial patterns of occupation[29] that form the basis for new, violence-free identities.

Finally, occupational therapists contribute to reducing and managing violence through promoting successful occupational performance, which may be defined as the 'choosing, organizing and carrying out… [of] occupations in interaction with the environment'.[23] Occupational performance involves a transactive and dynamic interplay between a person's abilities, their environment, and the demands of their chosen occupation.[31] Ultimately, this interplay either facilitates or inhibits an individual from completing activities and occupations successfully.[31]

This transactive understanding can be applied to formulate violence within psychiatric inpatient settings. For example, an individual may be unable to progress to unescorted community leave and require an escort (the environmental factor) due to lacking emotional control (the intrapersonal factor) and therefore being verbally aggressive towards another person who is delaying them in a shop, instead of showing patience and consideration for others (the demands of the occupation). Thus, violence can be a direct consequence of a poor fit between a person's abilities, their environment and the demands of their chosen occupation. Precedent exists for applying this transactive approach to violence risk assessment. Indeed, Rai[32] and Robbins[33] have demonstrated the relationship between impaired occupational performance and violence in the workplace and schools respectively. Both highlight the environmental and interpersonal factors that occupational therapists must treat in order to reduce violence, with Robbins[33] explicitly mentioning the 'goodness of fit' that is necessary between these factors. This understanding aligns with the biopsychosocial approach to violence that is prominent in current thinking. Nevertheless, it goes one step further by adding occupation to the equation, thus cementing the role of occupational therapists in preventing and managing violence.

In summary, the wider literature suggests an a priori model which explains how occupational therapists play a vital role in reducing and managing violence among mental health inpatients. Specifically, this model includes occupational therapists' role in: (1) decreasing the presence of risk factors and increasing the presence of protective factors, (2) reshaping patterns of antisocial occupation and the identities these create and (3) reducing occupational dysfunction that results in violence.

## STAGE ONE: IDENTIFYING THE RESEARCH QUESTION
### Study rationale
Despite this tentative model, the actual contribution of occupational therapy to managing and reducing violence within mental health settings is currently ill-defined. Clarifying this contribution is essential if occupational therapists are to include violence reduction as a specific part of their efforts to improve patients' performance of—and participation in—essential and meaningful occupations, and thus promote their recovery from mental illness. Additionally, such efforts will have benefits beyond improving the health and well-being of those who commit violence. By reducing violence, occupational therapists may indirectly improve the health outcomes of its victims, reduce the use of restrictive practices, and lessen the financial costs of violence for mental health services.

Understanding the role of occupational therapy in this respect requires an overview of relevant, existing literature. This is best achieved through a scoping review, since such a review both maps existing knowledge and identifies gaps in this knowledge to inform future research priorities.[34] This review protocol is based on guidance from the Joanna Briggs Institute.[35] It also incorporates five of the six stages of a scoping review (as developed by Arksey and O'Malley[34] and subsequently enhanced by Levac et al[36]) and guidance from the Preferred Reporting Items for Systematic Reviews and Meta-Analyses (PRISMA) Extension for Scoping Reviews checklist.[37]

To prevent unnecessary duplication, a preliminary search for existing scoping and systematic reviews on the subject of occupational therapy and violence among mental health inpatients was carried out on 12 September 2020. The terms 'occupational therapy' AND 'violence', and 'occupational therapy' AND 'violence' AND 'mental health' were used to search the Joanna Briggs Institute (JBI) Evidence Synthesis, the Cochrane Database of Systematic Reviews, OTSeeker, PROSPERO, Google Scholar, CINAHL Plus, PsycINFO and AMED. No relevant reviews were identified.

### Patient and public involvement
Unfortunately, it has not been possible to involve either the public or patients in the design of the proposed review.

### Study aim and objectives
The aim of the review is to understand the role of occupational therapy in reducing and managing violence among mental health inpatients, and in so doing, to inform future research priorities. This aim will be achieved by addressing the following objectives:
1. to investigate how occupational therapy assessment contributes to predicting the risk of violence among mental health inpatients,

2. to explore how occupational therapy interventions can be used to decrease violence risk in inpatient mental health settings.

## STAGE TWO: IDENTIFYING RELEVANT STUDIES
### Inclusion criteria
#### Population
As highlighted earlier in this paper, violence is a common occurrence in all mental health inpatient settings. Therefore, this review will include papers that focus on mental health inpatients irrespective of age, diagnosis, gender, civil or forensic status and legal or voluntary status. Papers focussing on mental health outpatients will also be included. This is justified by the fact that relevant literature is anticipated to be limited and additional studies conducted within community settings may provide further information that addresses the study aim and objectives.

#### Concept
Studies that apply any aspect of an occupational therapist's unique clinical reasoning to the assessment, formulation, reduction or management of violence committed by mental health inpatients will be included in this review. Essentially, relevant studies will seek to understand and/or treat violence from either an occupational therapy, or an occupational science, perspective. Studies which focus on discrete elements of function (eg, emotional regulation) in relation to violence, but do not do so from an occupational perspective will be excluded. Additionally, self-injurious behaviours will be excluded since they fall outside the HCR-20 definition of violence.[1]

#### Context
Since this study aims to map all available knowledge about the role of occupational therapy in reducing and managing violence among mental health inpatients, no exclusion criteria will be applied based on studies' context. Indeed, the authors will include studies irrespective of context-specific factors, for example country of origin, system of healthcare or specific mental health inpatient setting.

#### Types of studies
Studies will not be excluded based on year of publication, type of research, research design or quality. Research from every level of the evidence-based hierarchy[38] will be included, provided it meets the inclusion criteria for study population and concept. Relevant studies include those that are peer-reviewed and published within formal academic journals, as well as grey literature, including, but not limited to, conference presentations, policy documents and personal communications. Only studies written or available in English will be included.

#### Search strategy
In line with JBI guidance,[35] this review will adopt a three-step search strategy. First, the authors will search CINAHL Plus and PsycINFO for relevant studies; see online supplemental appendix 1 for the initial search strategy in CINAHL Plus. Since the reduction and management of violence relies on the expertise of various mental health professionals, relevant articles may be published in journals with a multidisciplinary readership, or in journals aimed specifically at occupational therapists. These two databases index articles from a wide range of disciplines related to mental health, and have been chosen for their breadth and scope.

Both authors will analyse the keywords in the titles and abstracts of all retrieved papers, as well as the index terms assigned to each study. They will incorporate these terms into the original list of search terms to produce a final search strategy. In the second stage, this comprehensive strategy will be tailored to each of the following databases: CINAHL Plus, PsycINFO, Medline, PsycARTICLES, ProQuest Health and Medicine, and AMED. Supplementary searching will be conducted in Google Scholar.

Following this search of the literature, the reference lists of all studies that will be included in the final review will be searched to identify additional relevant papers. This stage will also involve citation tracking and searches of grey literature. Grey literature searching will follow a three-stage process. First, searches for relevant theses will be conducted in the following resources: ProQuest Dissertations and Theses Global, DART-Europe E-theses Portal, Global ETD Search, Microsoft Academic Search, Bielefeld Academic Search Engine and EThOS e-theses online service. These resources have been selected for the breadth of literature that they index collectively. Second, conference proceedings will be located. This will involve searching www.opengrey.eu and the webpages of key conferences focussing on violence and mental health. Additionally, wherever possible conference proceedings will be located in the databases used for the main literature search strategy. Finally, the authors of relevant papers (identified either through searches of academic or grey literature) will be contacted to identify further resources.

## STAGE THREE: STUDY SELECTION
### Selection process
The results of the literature search will be exported and collated in the Rayyan web app,[39] and both authors will determine eligibility for each study. Initially, they will assess each result against the inclusion criteria using title/abstract alone. They will both do so independently and resolve any disagreements through consensus. Second, both authors will review the full-text of all articles selected for inclusion and determine eligibility against the inclusion criteria. Again, this will be done independently, and any disagreements will be resolved via consensus. The decision-making process of both authors will be made transparent through the inclusion in the final review of a PRISMA-ScR diagram,[37] and a narrative description of the literature search and study selection processes. Both methods will detail reasons for the exclusion of full-text articles.

To promote consensus, the authors will pilot study selection using the process suggested by the JBI.[35] Specifically, 25 random abstracts will be screened independently by each author using the review's inclusion criteria. The authors will then discuss the decisions made during this screening and amend the inclusion criteria as necessary. Screening studies for inclusion will only start when both authors have reached a consensus in 75% of cases or more.[35]

### Critical appraisal

Including a quality assessment in the proposed review risks divorcing the future research agenda from the realities of clinical practice and the lived experience of psychiatric inpatients and their families. Indeed, an evidence-based hierarchy will minimise 'poorer' quality evidence (eg, Eopinion pieces or editorials) and thereby dismiss the concerns of key stakeholders that they contain. Ultimately such an approach may prevent services from being shaped by those who need them, and by those who are responsible for delivering them.

Such concerns may account for various guidelines omitting critical appraisal as a requirement for a scoping review (see, eg, Khalil *et al*[40]). Nevertheless, critical appraisal can be considered a mark of quality[41] and necessary to ensure that scoping reviews provide meaningful evidence for policymaking, practice and research.[42] This review will appraise the quality of included papers using a simple checklist designed by Verhage and Boels[43] for use with quantitative, qualitative and mixed methods studies. Poorer quality evidence will not be excluded; however, the extent to which it influences and supports the review's findings will be identified using a sensitivity analysis.[43]

### STAGE FOUR: CHARTING THE DATA

A specific data-charting tool has been developed for the review, based on the tool provided in the JBI guidance for scoping reviews.[35] Charted data will include the following where relevant: author, location of study, date of publication, type of study, aims/purpose, sample size, gender of participants, participants' diagnoses, type of mental health inpatient setting (eg, medium secure forensic unit, or psychiatric intensive care unit), which aspect of occupational therapy or occupational science theory is included, which element of treating violence (ie, assessment/formulation/intervention/evaluation) is included, which aspect of the a priori model (explained in the introduction to this protocol) each study relates to, and key findings relevant to the review's aim and objectives.

To promote rigour, both authors will independently chart data for each study included in the review. The results of data charting will be discussed, and any disagreements resolved through consensus. As with the process of study selection, an initial pilot of data charting will be carried out using five studies. This will minimise any changes to the data-charting tool at a later stage.

### STAGE FIVE: SUMMARISING AND REPORTING RESULTS

Findings will be analysed and presented using both qualitative and quantitative methods. Specifically, quantitative data will be analysed using frequency counts according to the categories presented above, and presented in tabular format, with key data also being presented in bar and/or pie charts. Qualitative data will be analysed using Carroll *et al*'s[44] 'best-fit' framework synthesis. Data will be coded to the pre-existing themes identified within the chosen a priori framework drawn from the wider risk management and occupational therapy literature. Additional data that do not map onto these existing themes will be analysed using secondary thematic analysis.[45]

Both authors will perform this analysis independently, before reaching a consensus on which (if any) of the a priori themes, and which of the additional themes arising from secondary analysis are supported by the data. If sufficient data are available, all these themes will be used to construct a revised model explaining the role of occupational therapy in reducing and managing violence among mental health inpatients.

### ETHICS AND DISSEMINATION

The authors will disseminate the findings of this review by submitting it for publication in a peer-reviewed academic journal and via professional conferences. Since the review forms the foundation for additional research, additional dissemination may occur through the review being referenced in future academic papers. The review itself will deal exclusively with secondary data and therefore will not require ethical approval.

**Contributors** Both DB and NH were involved in designing the protocol. DB prepared the manuscript for publication.

**Funding** This work was completed as part of The Health Education England and National Institute for Health Research Integrated Clinical Academic Programme. Grant/award number: not applicable.

**Competing interests** None declared.

**Patient consent for publication** Not required.

**Provenance and peer review** Not commissioned; externally peer reviewed.

**ORCID iD**
David Bell http://orcid.org/0000-0002-8693-0537

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
