## [Reviewer comments · BMJ Open]

ARTICLE DETAILS

TITLE (PROVISIONAL)	The Role of Occupational Therapy in Reducing and Managing Violence amongst Mental Health Inpatients: A Scoping Review Protocol
AUTHORS	Bell, David; Hallett, Nutmeg

VERSION 1 – REVIEW

REVIEWER	Matthew Hirschtritt UCSF, Department of Psychiatry
REVIEW RETURNED	15-Dec-2020

GENERAL COMMENTS	The authors have proposed an important and well designed scoping review. I look forward to reading the results of their study.
--

REVIEWER	Dennis Grevenstein University of Heidelberg, Psychological Institute
REVIEW RETURNED	26-Dec-2020

GENERAL COMMENTS	The authors present a study protocol about a planned review on the influence of occupational therapy on violent behavior of psychiatric patients. The core idea is that occupational therapy provides key measures that could help prevent violence. The study protocol offers a reasonable theoretical background, promising methods and substantial scientific rigor. I am unfamiliar with the process of publishing a study protocol ahead of conducting a review, but if this is within the journal's policy I won't argue. The main issue I would have is that the study procedures are so vague that it is unclear if the authors could even be held accountable after completing their review. The reason would not even be the authors' fault, but rather be the unpredictability in this specific case and this specific subject. Due to the nature of the subject and the availability of respective studies, no specific statistical analyses will likely be necessary. Should the authors aim for specific qualifying statements, the authors should aim to quantify what data they have based their conclusions on. In any way, here are some general thoughts that might help sharpen the authors' view: 1: a priori model: Make this more specific. Highlight what exact pathways you predict and what exact aspects of occupational therapy should be able to cause what effect. Obviously, the empirical results will reject quite a bit of these predictions, but the process will help in assessing the results. Due to the lack of prior research, there is no clear expectation of the results, so it will be easy and convenient to rationalize any outcome as being within
---

	predictions. Having a clear a-priori model will counterbalance this mechanism. 2: specificity: One of the core themes will be to argue that some aspect or some effect is uniquely tied to occupational therapy. Granted, there is a certain subset of psychiatry focusing solely on bio-medical issues, but that is likely not true if you are looking for broader theoretical understanding. The biopsychosocial model leaves room for many different things and it might be difficult to specifically highlight the unique contribution of occupational therapy. For example, one could argue if a certain point is a unique quality of occupational therapy or rather a more general psychological process that just happens to be implemented in some form of occupational therapy, but might just as well be included in other therapy offerings. Even if that were the case occupational therapy might still be the best (most efficacious, most efficient, easiest...) way to implement it. Still, proper contextualization is advised. 3: context: Even though no specific hypotheses have been formulated, context likely matters when it comes to violence. It may be that occupational therapy does not universally work in all contexts to the same extent. 4: time: Violent behavior in inpatients usually does not occur randomly. Violence often comes in times of extreme exacerbation. Common psychiatric procedures most often preclude participation in occupational therapy for such patients. Results of the review should clearly delineate what point in time is focused. 5: make sure you seriously highlight limitations, unclear results and open questions. This will help guide future research in this under-researched topic. 6: stage six: discussing the results of the review in focus groups: I am really not sure what to expect here. I would probably neglect this part completely, unless the authors have a specific goal connected to these discussions, i.e. discussing a specific method or program that could be implemented and researched in the future. Anything else will lead to a confusion of audiences. Reviews are conducted and written for scientific audiences. If the authors desire, they can then translate some results or conclusions for other audiences, in order to get their feedback for future research. Getting feedback is not science in itself.
--	--

REVIEWER	Alaa Yousef University of Toronto, Institute of Medical Sciences (IMS)
REVIEW RETURNED	10-May-2021

GENERAL COMMENTS	Dear Authors, I appreciate the opportunity to review this solid scoping review. The proposal is clear and I have no comments at this stage.
---

VERSION 1 – AUTHOR RESPONSE

Reviewer 1 feedback

Dr. Matthew Hirschtritt, UCSF
Comments to the Author:

The authors have proposed an important and well designed scoping review. I look forward to reading the results of their study.

Authors' response

We would like to thank Dr Hirschtritt for the time and effort he has taken to review our paper.

Reviewer 2 feedback

Dr. Dennis Grevenstein, University of Heidelberg

Comments to the Author:

The authors present a study protocol about a planned review on the influence of occupational therapy on violent behavior of psychiatric patients. The core idea is that occupational therapy provides key measures that could help prevent violence.

The study protocol offers a reasonable theoretical background, promising methods and substantial scientific rigor. I am unfamiliar with the process of publishing a study protocol ahead of conducting a review, but if this is within the journal's policy I won't argue. The main issue I would have is that the study procedures are so vague that it is unclear if the authors could even be held accountable after completing their review. The reason would not even be the authors' fault, but rather be the unpredictability in this specific case and this specific subject. Due to the nature of the subject and the availability of respective studies, no specific statistical analyses will likely be necessary. Should the authors aim for specific qualifying statements, the authors should aim to quantify what data they have based their conclusions on.

In any way, here are some general thoughts that might help sharpen the authors' view:

1: a priori model: Make this more specific. Highlight what exact pathways you predict and what exact aspects of occupational therapy should be able to cause what effect. Obviously, the empirical results will reject quite a bit of these predictions, but the process will help in assessing the results. Due to the lack of prior research, there is no clear expectation of the results, so it will be easy and convenient to rationalize any outcome as being within predictions. Having a clear a-priori model will counterbalance this mechanism.

2: specificity: One of the core themes will be to argue that some aspect or some effect is uniquely tied to occupational therapy. Granted, there is a certain subset of psychiatry focusing solely on bio-medical issues, but that is likely not true if you are looking for broader theoretical understanding. The biopsychosocial model leaves room for many different things and it might be difficult to specifically highlight the unique contribution of occupational therapy. For example, one could argue if a certain point is a unique quality of occupational therapy or rather a more general psychological process that just happens to be implemented in some form of occupational therapy, but might just as well be included in other therapy offerings. Even if that were the case occupational therapy might still be the best (most efficacious, most efficient, easiest...) way to implement it. Still, proper contextualization is advised.

3: context: Even though no specific hypotheses have been formulated, context likely matters when it comes to violence. It may be that occupational therapy does not universally work in all contexts to the same extent.

4: time: Violent behavior in inpatients usually does not occur randomly. Violence often comes in times of extreme exacerbation. Common psychiatric procedures most often preclude participation in occupational therapy for such patients. Results of the review should clearly delineate what point in time is focused.

5: make sure you seriously highlight limitations, unclear results and open questions. This will help guide future research in this under-researched topic.

6: stage six: discussing the results of the review in focus groups: I am really not sure what to expect here. I would probably neglect this part completely, unless the authors have a specific goal connected to these discussions, i.e. discussing a specific method or program that could be implemented and researched in the future. Anything else will lead to a confusion of audiences. Reviews are conducted and written for scientific audiences. If the authors desire, they can then translate some results or conclusions for other audiences, in order to get their feedback for future research. Getting feedback is not science in itself.

Authors' response

We would like to thank Dr Grevenstein for his detailed feedback and are grateful for his efforts in reviewing our paper. We offer the following responses to his helpful comments.

Regarding the suggested vagueness of the study procedures, we have adopted a methodological framework that draws heavily upon established guidance from a number of sources, including the Joanna Briggs' Institute, Arksey and O'Malley's scoping review framework (updated by Levac, Colquhoun and O'Brien) and the PRISMA Extension for Scoping Reviews checklist. A number of recent protocols for scoping reviews published within BMJ Open have used similar methodological frameworks (see for example Mehotra S, Rowland M, Zhang H, et al. Scoping review protocol: is there a role for physical activity interventions in the treatment pathway of bladder cancer? *BMJ Open* 2019;9:e033518. doi:10.1136/bmjopen-2019-033518; Westerdahl F, Carlson E, Wennick A, et al. Teaching strategies and outcome assessments targeting critical thinking in bachelor nursing students: a scoping review protocol. *BMJ Open* 2020;10:e033214. doi:10.1136/bmjopen-2019-033214; and Fiore M, Bogossian E, Creteur J, et al. Role of brain tissue oxygenation (PbtO₂) in the management of subarachnoid haemorrhage: a scoping review protocol. *BMJ Open* 2020;10:e035521. doi:10.1136/bmjopen-2019-035521). We accept the unpredictability of both the results and the availability of relevant evidence, however we do not believe that this is influenced by the methods, which will provide transparency regarding the research process.

1: a priori model. The a priori model has been drawn from wider occupational therapy literature to focus and guide the process of mapping out and summarising existing knowledge regarding the role of occupational therapy in reducing violence, rather than to assist with making specific predictions on this subject. The purpose of the scoping review is not to offer statistical evidence of the effectiveness of occupational therapy in reducing or managing violence amongst mental health inpatients, but rather to provide an overview of literature pertaining to this subject and to inform future research. A possible outcome of the review may be the generation of specific hypotheses in relation to occupational therapy and violence reduction, and if this is the case, a future systematic review would be the appropriate method to test these.

2: specificity. We accept the challenges involved in attributing reduction in violence to the specific expertise of occupational therapists and acknowledge the presence of several confounding factors in this respect. However, the review does not seek to argue that any reduction in violence may be claimed exclusively by occupational therapists. Rather, it seeks to organise existing knowledge in this respect through refining a model based upon wider occupational therapy theory. It is anticipated that

the review will provide a hypothetical model which will assist in generating and testing hypotheses for future research, and that this future research may address the issues raised regarding the specific impact of occupational therapists' efforts to reduce violence. Nevertheless, reviewers' feedback on this point has made it clear that the third objective of the review – to examine the efficacy of occupational therapy in reducing and managing violence – is not congruent with the proposed methodology, and therefore this has been removed.

3: context. Consistent with the points above, hypotheses are not usually included in scoping reviews. Indeed, the latter are not required in guidelines from the Joanna Briggs' Institute or Arksey and O'Malley and are not contained within the Preferred Reporting Items for Systematic Reviews and Meta-Analyses Scoping Review checklist. For this reason, they are not included in the proposed review.

4: time. From our clinical experience in forensic mental health settings, we disagree with the statement that common psychiatric procedures often preclude participation in occupational therapy among patients who are extremely distressed. Occupational therapists are a core member of the multidisciplinary team within inpatient mental health settings and work with individuals irrespective of the degree of mental illness they are experiencing and/or the behavioural disturbances they may exhibit. Two examples may serve to illustrate this point. Firstly, occupational therapists work with patients in seclusion regularly in order to provide care and promote individuals' recovery irrespective of challenging behaviour. Secondly, many initiatives to reduce violence have their roots in sensory integration theory, which was developed by a prominent occupational therapist (Dr Jean Ayres). Given these facts, it is not necessary to restrict the focus of the review to any particular point in time.

5: limitations. We fully endorse the necessity to highlight limitations associated with the proposed review and have started this process by including a "strengths and limitations" section in the protocol abstract. More detailed discussion of limitations will be included in the final review, rather than in the protocol. Including all limitations in the final review will facilitate more comprehensive and meaningful assessment of rigour than would be the case if such discussion were split across two separate papers. This decision is consistent with JBI guidance for scoping review protocols (which does not recommend inclusion of limitations) and the recent BMJ Open scoping review protocols highlighted above.

6: focus groups. We fully accept the reviewers' comments in this respect and have removed this portion of the protocol.

Reviewer 3 feedback
Miss Alaa Youssef, University of Toronto
Comments to the Author:

Dear Authors,
I appreciate the opportunity to review this solid scoping review. The proposal is clear and I have no comments at this stage.

Authors' response
We are grateful to Miss Youssef and wish to thank her for reviewing our paper.

VERSION 2 – REVIEW

REVIEWER	Dennis Grevenstein University of Heidelberg, Psychological Institute
REVIEW RETURNED	01-Jul-2021

GENERAL COMMENTS

the authors present a revision of their study protocol about their planned review on occupational therapy and violence.

In my previous review, I described six different points that might be improved. After reading through the manuscript, I can only see that the authors resubmitted a largely unchanged paper, mostly with stage six being cut out completely. The only substantial thing added is a short paragraph about the importance of occupational therapy in psychiatry in general; something I never contested.

In their rebuttal letter, the authors describe that they essentially disagree with my comments and hence chose to ignore them. Okay. I just don't know what to do with such a revision.

As far as I see it, this review has open und unclear scope, almost no constraints, and no clear expectations about the outcomes. If that is what you want, please go ahead and do your review and see what you find. I just don't see the purpose of a study protocol here.